# The Use of Ceus Software with No Contrast Media Administration in the Diagnosis of Pneumoperitoneum

**DOI:** 10.3390/diagnostics12020401

**Published:** 2022-02-03

**Authors:** Michele Altiero, Giuseppina Dell’Aversano Orabona, Ettore Laccetti, Alessandro Rengo, Roberta Danzi, Federica Romano, Marco Di Serafino, Francesca Iacobellis, Giampiero Francica, Mariano Scaglione, Luigia Romano

**Affiliations:** 1Department of Diagnostic Imaging, Pineta Grande Hospital, 81030 Castel Volturno, Italy; michelealtiero@hotmail.it (M.A.); ettorelac@hotmail.it (E.L.); alessandrorengo@hotmail.it (A.R.); giampierofrancica@gmail.com (G.F.); scaglionefun@gmail.com (M.S.); 2Department of General and Emergency Radiology, “Antonio Cardarelli” Hospital, A. Cardarelli St. 9, 80131 Naples, Italy; marcodiserafino@hotmail.it (M.D.S.); iacobellisf@gmail.com (F.I.); luigiaromano1@gmail.com (L.R.); 3Department of Radiology, S. Maria delle Grazie Hospital, 80078 Pozzuoli, Italy; danziroberta@gmail.com; 4Department of Radiology, Monaldi Hospital, 80131 Naples, Italy; fedinromano@gmail.com; 5Department of Radiology, University of Sassari, James Cook University Hospital, Marton Road, Middlesbrough TS4 3BW, UK

**Keywords:** acute abdomen, pneumoperitoneum, gastrointestinal perforations, emergency, contrast-enhanced US examination (CEUS)

## Abstract

Background: Pneumoperitoneum is defined by the presence of free air in the abdominal cavity; gastrointestinal perforation is an important cause of this pathological condition. In emergency situations, radiology is considered vital in the early detection and identification of the site and cause of the perforation, which is critical for proper surgical planning. Aim: The aim of our study was to evaluate a new diagnostic US tool, based on the US contrast-specific software generally used during contrast-enhanced US examination (CEUS), without the administration of sonographic contrast media, and to describe the specific imaging features in the detection of free intra-peritoneal air. Subjects and Methods: One hundred and fifty-seven consecutive and hemodynamically stable patients, who arrived in our E.D. with an acute abdomen between April 2018 and October 2019, underwent US and CT examination, performed by three radiologists (with 5, 5, and 25 years of experience). The US was performed first and divided into two steps, using B-mode US and both B-mode and contrast-specific software US, with no contrast media administration. All the patients underwent CT examination. Results: In 32 out of 157 patients, the surgery confirmed GI perforation. CT correctly detected 31 out of 32 patients; the contrast-specific software US identified 30 perforated patients. CT reached a sensitivity value of 97% and specificity value of 100%; contrast-specific software US demonstrated higher values than B-mode US in sensitivity (93% vs. 70%, respectively) and specificity (98% vs. 88%, respectively). Conclusion: the use of contrast-specific software in emergencies improves image quality, and reaches higher levels of sensitivity and specificity with no time delay compared to standard US examination, helping radiologists expedite diagnoses.

## 1. Introduction

Pneumoperitoneum (PNP) is a common clinical entity defined by the presence of free air or gas in the peritoneal cavity [1]. Gastrointestinal (GI) tract perforations represent an important cause of PNP, a potentially life-threatening condition associated with high morbidity and mortality (30–50%), with even poorer outcomes when the diagnosis is delayed [2,3]. As documented in the previous literature, GI tract perforations can have various causes, including trauma and iatrogenic injury, inflammatory conditions, infection, ischemic change, diverticula, foreign bodies and malignancy [4,5]. It is important to correctly identify the site and the cause of the perforation in order to appropriately manage and make decisions regarding surgical planning. However, clinical diagnosis of the GI perforation site may be difficult, as symptoms may be non-specific (abdominal pain and distension, vomiting, constipation, fever, diarrhea, tachycardia, hypotension, and tachypnea) and related to multiple factors, including the source of the perforation and its mechanism, time elapsed since the perforation, the degree of contamination of the peritoneal cavity and the patient’s age [1]. In the emergency setting, radiology is vital in the early detection of this pathologic condition [6].

Until now, the abdominal upright posteroanterior X-ray is traditionally regarded as the foremost method of detection, with the relevant signs described in [7,8].

However, extraluminal air is not always demonstrable on plain abdominal radiography, especially if self-sealed, well contained by adjacent organs, or if the perforation is too early or small, considering that sensitivity collapses (50–70%) for GI perforation with less than 1 mL of gas [9]. The patient’s clinical status could also make the recognition of free air difficult, reducing the method’s diagnostic value in patients too sick or debilitated to stand up for an erect abdominal plain film [8]. Only 55–85% of PNP could be detected by abdominal X-ray plain film [8].

Computed tomography (CT) is the modality of choice in the detection of GI tract perforation, due to its high sensitivity and accuracy (82 to 90%); it can display intra- and extra-peritoneal air, localizing the perforation site [10]. CT demonstration of PNP occurs when free gas, discontinuity of the GI wall or leakage of orally administered contrast medium are visible [11]. Other signs may include fluid abdominal effusion [12], inhomogeneous mesentery, wall thickening, “dirty mass” (extra-luminal fecal matter) [13], intestinal or porto-mesenteric pneumatosis, and abdominal abscess [8,14]. A total of 83–100% of PNP could be diagnosed through CT [6], although it is not cost effective and is associated with radiation exposure.

Ultrasound (US) findings of PNP were first identified in 1984 [15]. It has been profusely demonstrated that US could be useful in identifying PNP and investigating its causes [6,9], as reported in several studies and case reports [16,17,18], attesting to very different values of diagnostic accuracy (53–100%) [19,20].

In free abdominal air detection with US, the use of the linear array transducers (10–12 MHz) represents the most sensitive standard of detection, thanks to its high resolution [21]. The supine position with the thorax slightly elevated (10–20 degrees), and the prone position exploring the right paramedian epigastric area, are the best methods, while the right upper quadrant and the pre-hepatic space are the most common sites of air accumulation [22].

The US detects signs of free abdominal air via the scattering of US waves in correspondence with the anatomic interfaces between soft tissue and bubble gas, associated with reverberation phenomena of the waves between the transducer and air bubbles. This event produces an increased echogenicity of a peritoneal stripe accompanied by numerous reflection artefacts, typically with a ring-down or comet-tail appearance, associated with the characteristic feature of real-time modifications as the patient’s position changes [6,21,22,23].

The diagnostic confidence in US rises when increased thickness of the bowel wall, collected fluid effusion and a reduction in peristaltic moves are also present [24]; on the other hand, intraluminal air may be recognizable when peristalsis and normal wall thickness are present. The detection of peritoneal stripes, especially if modified by the patient’s position change, is very suspect for PNP [25]. In addition, US permits one to observe the motion of free air in real time, to discriminate the air in the lungs from PNP, thanks to respiration excursion [22], to localize the presence of retroperitoneal perforation detected by air around the duodenum and the pancreatic head [26], and to use specific US methods, such as the scissor maneuver by Karahan [27].

Moreover, US is widely recognized as an indispensable tool in the bedside diagnosis of the acute abdomen, and in the trauma context (FAST), where it has been recommended as standard procedure [28]. It may also be useful in patients where radiation burden should be limited, for example, children and pregnant women [6]. When compared to plain radiography, US demonstrates greater sensitivity (93% vs. 79%, respectively), very similar specificity (64%), and a positive predictive value (97%) [29].

Despite this, many text books and lecturers, especially in the field of emergency and critical care, skip the topic entirely. US has not been adequately integrated into the standard diagnostic process of PNP detection. The usual explanation for this is that gas is a strong reflector, able to prevent the transmission of US waves and create reverberation artifacts, inhibiting the obtainment of some diagnostic information. In addition, the physiological gas within the bowel may make it even more difficult to obtain an accurate interpretation [24].

The aim of this study was to evaluate the usefulness of a new diagnostic US tool in PNP detection, based on the imaging performed by the US contrast-specific software, which is the software generally used during contrast-enhanced US examination (CEUS). In our series of tests, the contrast-specific software was routinely used in the emergency setting as the first tool to assess any patient with an acute abdomen. This was performed in an innovative way, as it was not dependent on the administration of contrast agents, and was defined as contrast-specific mode (C-mode). The C-mode operates in real-time, with a low mechanical index (pulse inversion technology), and uses a digital subtraction in order to isolate the signal in double harmonic. [30] Thanks to this method, the receiver electronically filters and deletes the fundamental frequency, showing only the double harmonic signal on the monitor, which generally originates from the vessels and from the air bubbles in the abdominal cavity in our specific case. Our intent is to help the radiologist diagnose abdominal free air with confidence, implementing US B-mode evaluation through the use of C-mode, which strongly emphasizes the sonographic air signal.

## 2. Methods

Our study included 157 consecutive patients who arrived in our emergency department with acute abdomen between April 2018 and October 2019. Hemodynamic instability and renal failure were considered exclusion criteria. All the included patients underwent US examination first, divided into two steps, and then a CT examination (Figure 1). This workflow did not cause any significant time delay in the diagnosis and management decision process.

The US examination was performed using a Resona 7 system (Shenzhen Mindray Bio-Medical Electronic Co, Shenzhen, China), equipped with both curved- and linear-array probes, and with dedicated software for acquisition of contrast-specific imaging. No administration of US contrast media was used.

All CT examinations were performed with a 128-slice DSCT (Somatom Definition Flash; Siemens Healthcare, Forchheim, Germany) after iodinated contrast media administration (Iomeprol injectable solution, Iomeron 400); images were acquired in a single portal venous phase.

The US examination was performed by two different radiologists with no less than 5 years of experience in the emergency department (R1 and R2); CT evaluation was performed by a radiologist with 25 years of experience (R3).

The US examination was divided into two steps, consisting, first, of an examination performed using baseline US (B-mode), and a second examination using the mentioned US technology consisting of C-mode, with no contrast media administration, to focus on the free air detection. R1 and R2 conducted the two steps of the US examination for each patient; the interobserver concordance rate was calculated to explore the diagnostic performance and reproducibility.

All patients underwent CT examination after the US study was performed by R1 and R2 and the CT was examined by R3. Radiologists were blinded to the results of the other examinations. Since CT is considered as the gold standard in free air detection, the statistical analysis allowed us to estimate the sensitivity, specificity and diagnostic accuracy for CT, US B-mode and C-mode. Positive and negative predictive values of US B-mode and C-mode were also evaluated. Written informed consent was obtained from each patient.

## 3. Results

Our population of study included 157 patients with an acute abdomen (96 M and 61 F; mean age 41 ± 16 years). The main characteristics are described in Table 1.

In this study, 32 out of 157 patients (20%) received surgical confirmation of GI perforation. The other causes of an acute abdomen were acute diverticulitis (*n* = 39), intestinal occlusion (*n* = 24), acute appendicitis (*n* = 18), acute pancreatitis (*n* = 14), acute cholecystitis (*n* = 12), ischemic colitis (*n* = 5), renal colic (*n* = 5), intestinal volvulus (*n* = 3), inflammatory bowel disease (*n* = 3), and ovarian torsion (*n* = 2).

A statistical analysis of the results was conducted in order to calculate the sensitivity, specificity, diagnostic accuracy, and positive and negative predictive value of the different methods. Therefore, the concordance index using the Choen’s *k* test for C-mode and B-mode in the detection of free air was calculated.

In our population, CT correctly detected 31 out of 32 patients with GI perforation.

The CT value of sensitivity and specificity in free air detection were 97% and 100%, respectively, with a high level of diagnostic accuracy (99%) in showing free air.

In the analysis results of both R1 and R2, C-mode US demonstrated higher average values than B-mode US in sensitivity (93% vs. 70%, respectively), specificity (98% vs. 88%, respectively) and diagnostic accuracy (97% vs. 81%, respectively). C-mode US identified 30 perforated patients in R1’s examinations and 29 in R2’s examinations, while B-mode reached the diagnosis in 23 and 21 cases, respectively, in R1 and R2’s examinations.

B-mode US examinations performed by R1 erroneously suspected the presence of PNP in 9 cases (6% false positive), while R2 obtained 10 false positives (6.4%). C-mode US obtained only two false positives (1.3%) from R1 and three false positives (1.9%) from R2.

In no case did any CTs performed by R3 result in a false positive.

On the other hand, the false negative average value for B-mode and C-mode was, respectively, around 6% and 1.6%, while the false negative rate for CT examinations was 0.6% (1 case). Therefore, the negative predictive average value was 98% for C-mode US and 93% for B-mode US, while the positive predictive value was 94% for C-mode US and 71% for B-mode US.

See Table 2.

The interobserver concordance index for C-mode between R1 and R2 was 99% (Cohen’s *k*: 0.979); the same index for B-mode was 99% (Cohen’s *k*: 0.946). The R1 intraobserver concordance index for C-mode and B-mode detection of PNP was 77%, and the R2 intraobserver concordance index for C-mode and B-mode detection of PNP was 74%, but the Choen’s *k* results were unsatisfactory in both (0.175 and 0.253, respectively), confirming the limited validity of B-mode US as a diagnostic test for air detection.

Concerning the localization of free air bubbles, in our cases, the preferred sites to exalt the presence of free abdominal air were found to be the interface with the anterior abdominal wall, hepatic interspace (peri-hepatic profiles, falciform ligament and hepatic hilum cavity), and peri-duodenal space.

## 4. Discussion

The rationale of our study was to utilize the strengths of US in the assessment of PNP, and demonstrate its diagnostic power without the use of contrast media; to date, we have not found similar studies in the literature.

In most European emergency departments, US is routinely performed by radiologists in their first assessment of any patient with an acute abdomen [31].

Today, the majority of modern US machines are already equipped with specific software for CEUS, with a large possibility of scan acquisition and co-registration after the administration of contrast agents.

The evaluation of B-mode plus C-mode examination was important to understand if any significant diagnostic information was added, as confirmed in our results.

One patient, whose micro-perforated duodenal ulcer was found via surgery, could not be diagnosed by CT, probably due to the low amounts of sufficiently detectable free gas and the poor communication of indirect signs. In the same patient, C-mode and B-mode were also not able to provide the diagnosis for both US examinators.

As reported in our results, 30 patients were correctly detected on C-mode by R1. In two patients, C-mode was not able to reach the diagnosis of GI perforation; surgical intervention confirmed the presence of a micro-perforated duodenal ulcer and self-sealed peri-diverticular air, respectively. The C-mode performed by R2 identified 29 perforated patients, adding one more missed diagnosis of PNP in a case of peri-duodenum perforation in post-attinic parietal change due to pancreatic carcinoma history.

B-mode US was not able to correctly detect the presence of free abdominal air in 9 and 10 patients, respectively, missing the diagnosis.

Two cases of C-mode false positives were obtained from R1 and R2 examinations, in the first hypothesis, due to intraluminal bowel gas artefacts.

In our experience, the statistical results showed higher sensitivity and specificity in C-mode than in B-mode US (93% vs. 70% and 98% vs. 88%, respectively); taking CT into consideration as the gold standard, with a reported diagnostic accuracy rate of 99%, the results also confirm the high diagnostic accuracy of this innovative tool (97%) and its high negative predictive value (98%).

The high value of the inter-observer concordance index (99%) reflects well upon the reliability of the C-mode method, demonstrating its reproducibility.

The qualitative superiority of C-mode over B-mode is adequately demonstrated by the imaging.

In our patients, larger air bubbles appeared as bright, highly echogenic lines with distal reverberation and shadowing artifacts, as ring-down or comet-tail artifacts on B-mode

(Figure 2a); free air can also be detected beneath the anterior abdominal wall, where it generally accumulates in the supine patient (Figure 2b). 

C-mode gave a more intense appearance to the peritoneal stripe when compared to B-mode, identifying the amount of free air, similar to the strongly enhanced peritoneal lines in the pre-hepatic space (Figure 3), and better enounced the presence of intensely enhanced small amounts of air around the falciform ligament (Figure 4a), as confirmed by the CT scan (Figure 4b).

Therefore, in some cases of our study population, a small amount of air under the abdominal wall was not clearly visible, and was potentially lost when the examination was not performed in expert hands, where it was misunderstood as a simple anatomic interface on B-mode (Figure 5a,b). In these cases, C-mode demonstrated its ability to distinctly reveal highlighted peritoneal stripes (Figure 5c) that modified their aspects as the patient’s position changed (Figure 5d).

In another case, C-mode was able to detect free abdominal air around the duodenum and the pancreatic head with certainty (Figure 6a), which was also displayed perfectly by the CT scan (Figure 6b).

As observed in Figure 7, an accurate B-mode examination could also lose a minimal amount of free gas; the same patient was correctly detected on C-mode, discriminating a few air bubbles as bright punctuate foci of the luminescent echo line.

Moreover, in one obese patient, C-mode better revealed the presence of PNP and demonstrated its efficient performance, despite the thick fat plane in the anterior abdominal wall and the interposed meteorism, when compared with B-mode.

In addition, in one single case of a child with appendicitis complicated by perforation (excluded from the present study for the lack of CT examination), our diagnosis was confident, thanks to the clear integration of C-mode scans during the US examination; the diagnosis was then surgically confirmed.

As mentioned in our results, air bubbles were most frequently located in the anterior abdominal wall, hepatic interspace (peri-hepatic profiles, falciform ligament and hepatic hilum cavity), and peri-duodenal space. However, the presence of suvra-mesocolic free air does not exclude a sub-mesocolic site of leakage in the GI system. Moreover, one must consider the spontaneous trend of air to move upwards, according to the anti-gravity effect; CT examination results were mandatory in order to detect the leakage.

Although the localization of detected free air bubbles cannot be considered a reliable criteria of C-mode, to obtain deep information about the site of perforation, the benefits of C-mode are proved when compared to B-mode US, as shown in our study. GI perforation does not display specific symptoms, and abdominal X-rays only detect free air in a limited percentage of cases, dependent on the size and progress of the perforation; nevertheless, not all the patients with acute abdomen immediately undergo CT examination. Therefore, any patient who arrives at the E.D. with acute abdomen should receive a C-mode examination, so that PNP may be detected and diagnosed as soon as possible. Consequently, the decision to quickly and decisively utilize CT examination and treatment will save time and reduce the number of people with undiagnosed PNP.

## 5. Limitations

The limitations of our study are as follows: (1) The small population size of our test group. It will be necessary to apply the same method to a larger population to confirm our results; (3) False negatives. The number of false negatives is small in this study, but represents a limit to exceed in order to increase the reliability of C-mode’s diagnosis of PNP.

## 6. Conclusions

The use of C-mode US examination in the emergency department demonstrated its superior image quality, and its higher levels of sensitivity, specificity and diagnostic accuracy.

This method did not change the number of necessary scans and the timing of US examination. Furthermore, it has demonstrated that it can reliably diagnose PNP without the use of iodinated radiation or the administration of contrast agents.

In addition, this method’s performance has proven reliable, even when dealing with patients who would otherwise be difficult to diagnose, such as children and the obese. It may, therefore, be used instead of a CT examination, which may be held in reserve should abdominal pain remain undiagnosed after US examination.

In our experience, this new method could provide better visibility of the diagnostic elements of free abdominal air detection, resulting in increased confidence in the diagnosis of PNP; the C-mode US could help young radiologists, or radiologists with poor experience, in emergency clinical situations, to not miss this important diagnosis.

## Figures and Tables

**Figure 1 diagnostics-12-00401-f001:**
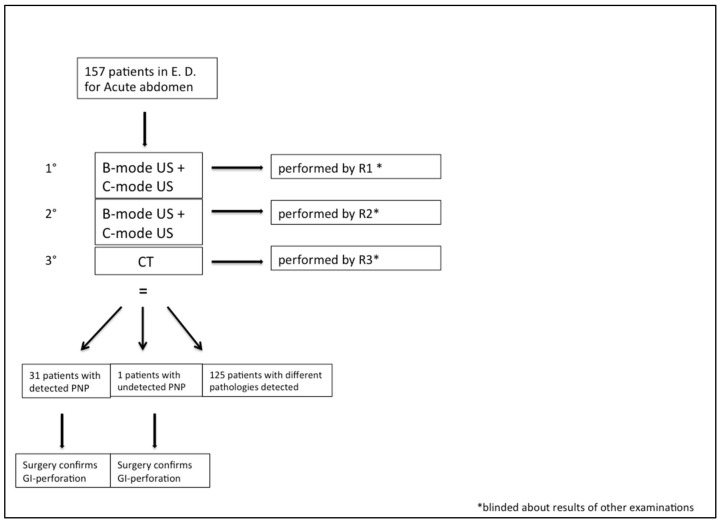
Study population design. According to the flowchart of the study procedure, all patients underwent B-mode, C-mode and CT examination, performed by three different radiologists (R1, R2, R3). At the end of the presented diagnostic course, PNP was correctly detected in 31 patients and missed in one patient. Successive surgical intervention confirmed the presence of PNP originated from gastro-intestinal perforation in 32 patients. * E.D. = emergency department. * R1, R2, R3 = radiologist 1, radiologist 2, radiologist 3. * C-mode = contrast-specific software mode. * PNP= pneumoperitoneum. 1°,2°,3° = first step, second step, third step of study.

**Figure 2 diagnostics-12-00401-f002:**
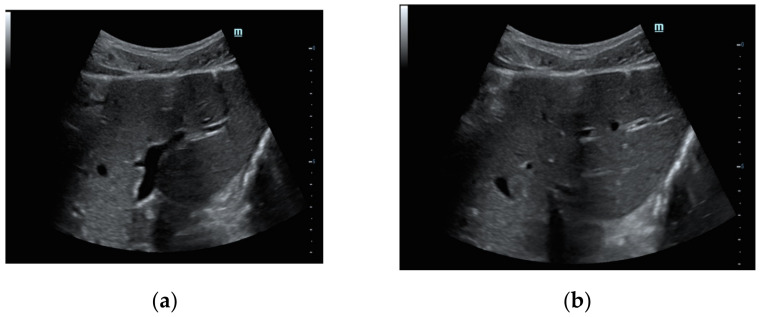
Female patient, 37 years old with E.R. access for acute abdomen and successive surgical confirmation of GI perforation. (**a**) Transverse midline US abdomen scans in B-mode show typical artefacts related to free bubble gas presence as bright and highly echogenic lines with distal reverberation and shadowing artifacts (comet-tail artifacts), especially in the pre-hepatic plane. (**b**) The detection of free air is typically found by looking under the anterior abdominal wall, considering the free air bubble’s tendency to be placed upwardly.

**Figure 3 diagnostics-12-00401-f003:**
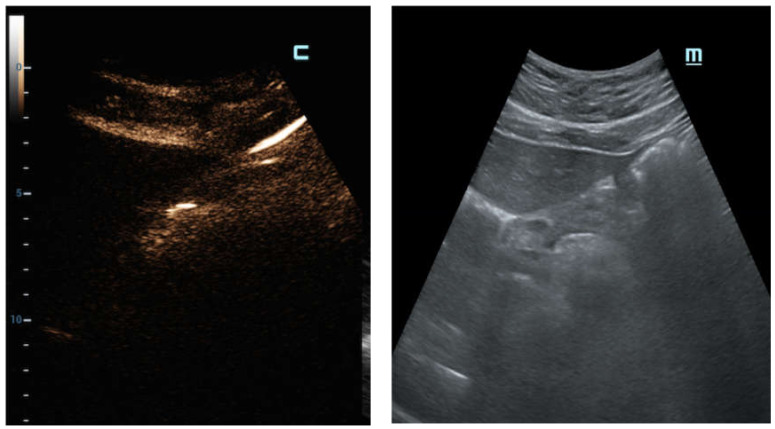
Male patient, 64 years old, surgically treated for GI perforations with finding of gastric foreign body (fish bone). Longitudinal US scan focused on the pre-hepatic space; the C-mode window is automatically displayed when the contrast-specific program is launched. C-mode clearly documents the presence of peritoneal stripes in the pre-hepatic space and around the liver. The visibility of these diagnostic elements is immediately distinguishable on C-mode’s window when compared to B-mode’s image.

**Figure 4 diagnostics-12-00401-f004:**
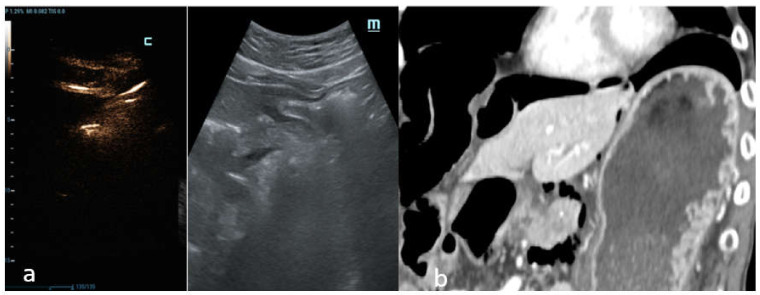
Male patient, 71 years old with story of analgesic abuse for myalgia in treatment. (**a**) Longitudinal US scan on the hepatic falciform ligament; the C-mode and B-mode image windows are displayed side by side. The C-mode shows the intensely bright appearance of a small amount of air placed along the falciform ligament; this discovery is explicit on C-mode and more evident than B-mode. (**b**) Coronal MPR reconstruction of abdominal CT in venous phase of the same patient demonstrates the presence of pneumoperitoneum with free air bubbles around the liver and the falciform ligament. Perforation in the first tract of duodenum was identified via laparotomy performed in the emergency department.

**Figure 5 diagnostics-12-00401-f005:**
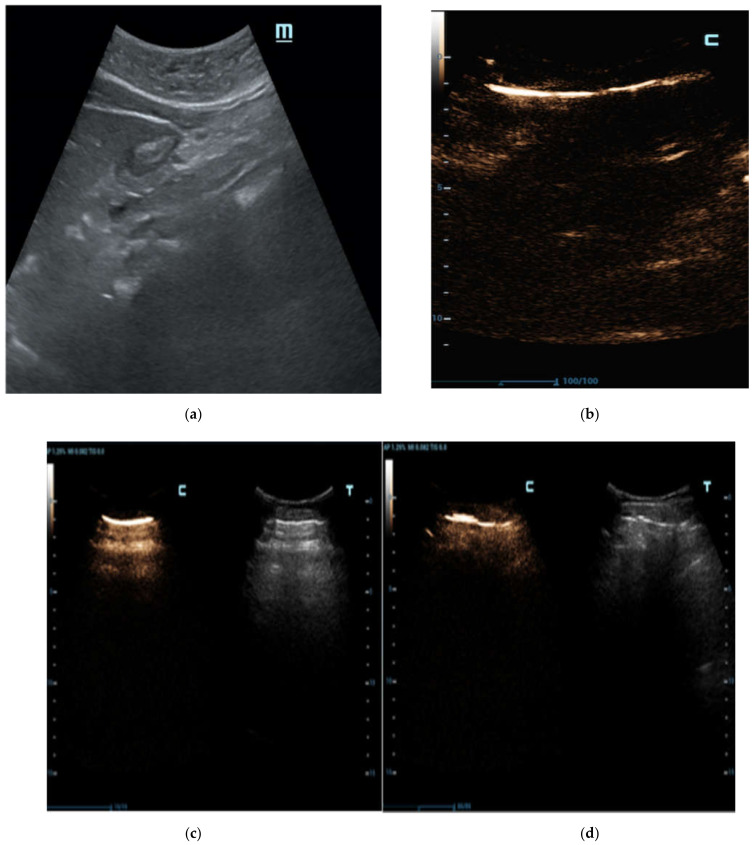
Black male patient, 29 years old with acute abdominal pain; no comorbidities were noted at the time of E.D. access and radiologic evaluation. (**a**) Longitudinal sub-hepatic US scan of liver. Air under the abdominal wall was erroneously interpreted as hyper-echogenicity in correspondence with B-mode’s anatomic interface; (**b**) C-mode demonstrated its ability to distinctly reveal highlighted lines in correspondence to the pre-hepatic space. (**c**) Transverse abdominal US scans show the peritoneal plane under anterior abdominal wall. C-mode distinctly reveals intensely highlighted lines of brightness under the abdominal wall; (**d**) the peritoneal stripes’ appearance is modified when the patient’s position is changed at the radiologist’s instruction.

**Figure 6 diagnostics-12-00401-f006:**
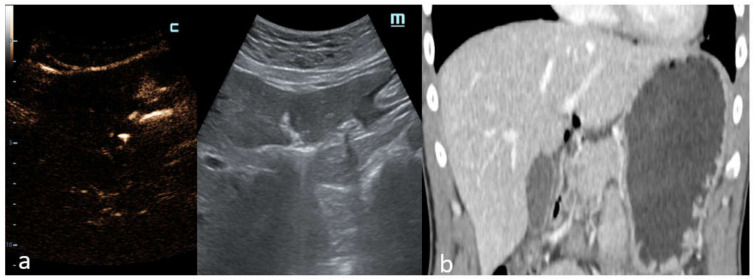
Male patient, 35 years of age with abdominal pain and a prolonged history of alcohol abuse. (**a**) Intercostal US scan of gallbladder region with depiction of C-mode and B-mode images. Free abdominal air around the duodenum and the pancreatic head is documented. (**b**) Coronal MPR reconstruction of abdominal CT in the venous phase shows increased thickness and focal discontinuity of the second tract of the duodenal wall; extra-luminal free air is also detected. Surgery revealed the presence of pyloric ulcerated perforation.

**Figure 7 diagnostics-12-00401-f007:**
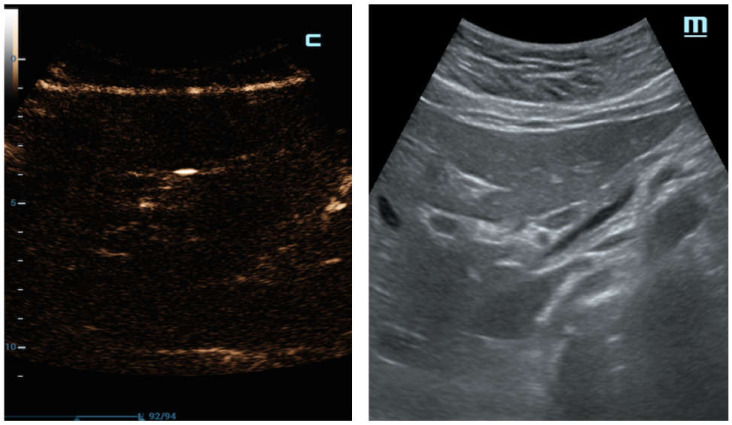
Female patient, 51 years old with history of radiotherapy and chemotherapy for pancreas neoplasm. Surgery revealed micro-perforation in the first tract of duodenum related to post-actinic damage in correspondence to the duodenum wall. Longitudinal US C-mode scan of the liver correctly detects the few air bubbles displayed as bright luminescent echo lines; this finding is clearly observed on the C-mode; as shown, it is very difficult to find clear corresponding data with B-mode.

**Table 1 diagnostics-12-00401-t001:** Population’s main characteristics.

Study Populations	*n* = 157
Sex	
M	*n* = 96
F	*n* = 61
Body weight (Kg)	61 ± 16.50
Age (years)	41 ± 16
Definitive diagnosis	
PNP from GI-perforation	*n* = 32
Acute diverticulitis	*n* = 39
Intestinal occlusion	*n* = 24
Acute appendicitis	*n* = 18
Acute pancreatitis	*n* = 14
Acute cholecystitis	*n* = 12
Renal colic	*n* = 5
Ischemic colitis	*n* = 5
Intestinal volvulus	*n* = 3
IBD	*n* = 3
Ovarian torsion	*n* = 2

IBD = inflammatory bowel disease. PNP = pneumoperitoneum.

**Table 2 diagnostics-12-00401-t002:** Statistical analysis of results.

	CT	C-MODE US	B-MODE US
**Sensitivity** **(%)**	97	93	70
**Specificity** **(%)**	100	98	88
**Diagnostic accuracy** (**%)**	99	97	81
**False positive (FP)** **(%)**	0.6	1.6	6.2
**Positive predictive Value (PPV)** **(%)**	-	94	71
**Negative predictive Value (NPV)** **(%)**	-	98	93

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
