# Peer review of "The Use of Ceus Software with No Contrast Media Administration in the Diagnosis of Pneumoperitoneum"

_diagnostics, 2022, doi:10.3390/diagnostics12020401_

Round 1
Reviewer 1 Report
It's an interesting and well written work. Please add your ethical approval in the methods session and check the english language in the abstract.
Author Response
As suggested, the ethical approval has been added in the Methods and the English language has been improved in the abstract and in the main text.
We would like to thank the Reviewer for taking the necessary time and expertise to review the manuscript. We appreciate each comment and suggestion.
Reviewer 2 Report
Comments to Authors
In this retrospective study authors present a rather new application of ultrasound with CEUS software in the emergency department without contrast media application for the detection of pneumomediastinum. They show that C-mode US has higher sensitivity and specificity than B-mode US in detecting free abdominal air in a study population of 157 patients, comparing it also to CT examination as gold standard. They conclude that this method may improve clinical practice.
The manuscript is of appropriate length. The single sections are well organized. Language, images (and figure legends), tables and references are concise and of sufficient quality.
DETAILED COMMENTS
TITLE
The title is informative and fairly attractive.
ABSTRACT.
- Lines 12-13, typo: First, when beginning a sentence with a number, please write out the number (one hundred fifty-seven). Second, please add “who” and comma in the sentence as follows:
“157 consecutive and hemodynamically stable patients, who arrived in our E.D. with 12 acute abdomen between April 2018 and October 2019, underwent US and CT examination…”
- Line 19, typo: “CT reached a value of…” not “to”. Please change.
KEYWORDS
- Keywords are missing completely. Please provide.
ABBREVIATIONS
- Also completely missing. Please give a list containing for instance US, CEUS, CT, PNP, etc..
INTRODUCTION
- The introduction is overall well organized, but can be shortened. For instance consider to summarize paragraphs beginning from line 67 to 81.
- Line 52, typo: Please delete “to” in “…and to localize…”.
- Line 60, typo: Delete “a” in “…attesting to a very…”
- Line 102, typo: “uses” instead of “use”.
- Please give references as to the use and mechanism of C-mode US (lines 101-105).
MATERIAL AND METHODS
- Line 115, typo: Please remove the hyphen in “with-an”.
- Line 142: Repetitive, please delete the last sentence, as the calculation of inter-observer concordance was mentioned previously.
RESULTS
- Line 147: Please insert n= before (24).
- Values of sensitivity and specificity for C-mode US vs B-mode US are not similar in Abstract and Results section: In the abstract it says: sensitivity 94 vs 69% and specificity 98 vs 92%, whereas in the results section numbers of 93% vs 70% for sensitivity and 98% vs 88% for specificity are given. Which are correct?
- Was it also possible to correctly detect the location of air leakage using C-mode or simply the presence of abdominal air? Could you provide data for both readers for this point?
DISCUSSION
- As I take it, even if C-mod US detected free abdominal air, a CT scan is mandatory in order to detect a possible leakage. Could you discuss the benefit of an additional, probably time consuming US examination in this respect?
FIGURES/TABLES
- Good images, however, I am missing the arrows described in the legends.
- Line 400, typo: “Correct “note” to “noted”.
BIBLIOGRAPHY
Okay.
INFORMED CONSENT/ ETHICAL CONSIDERATIONS
Missing.
Author Response
We sincerely would like to thank the Reviewer 2 for his thoughtful comments and efforts. We have taken the precious comments on board to improve and clarify our manuscript. Thank you for the expertise that you contributed towards reviewing the article.
Please see the attachment.

Reviewer 3 Report
This interesting study of 157 patients with acute abdomen sought to assess diagnostic value of a US contrast-specific software without administration of sonographic contrast media. Overall, the study is well-written and fairly well-composed. My comments are as following.
Major comments
- Why were patients with hemodynamic instability excluded from the study? These patients with acute abdomen represent a clinically significant group in which fast, bedside diagnosis is especially warranted.
- There is no statistical analysis of the results and the relevant section is missing. Did the authors compare statistically the diagnostic efficacy of the different methods? Was there a significant difference between the accuracies of the US methods in terms of PNP diagnosis. What was the agreement between the two methods for PNP diagnosis? If possible, statistical analysis should be provided including the statistical tests that were used in data analysis. The specific software used for statistical analyses should also be mentioned. If statistical analysis was not possible, this should be further explained.
- Is this the first study addressing the utility of this US contrast-specific software in the diagnosis of PNP or are there any relevant data found in the literature? This should be mentioned and analysed in the discussion section.
Minor comments
1. In my opinion, the introduction section seems somewhat long.
2. A table showing the sensitivity, specificity, PPV, NPV, and accuracy for each diagnostic method would be useful.
Author Response
We would like to thank the Reviewer for taking the necessary time and expertise to review the manuscript. We appreciate each comment and suggestion.
Major comments
- Why were patients with hemodynamic instability excluded from the study? These patients with acute abdomen represent a clinically significant group in which fast, bedside diagnosis is especially warranted.
Thank you for this important clarification. In case of hemodynamic instability the patient is firstly assisted with cardio-respiratory and fluid management and immediately elected for CT examination, in our Institutes. The possibility to apply the C-mode also in this case should be investigated and we could improve our next study in this way.
2.There is no statistical analysis of the results and the relevant section is missing. Did the authors compare statistically the diagnostic efficacy of the different methods? Was there a significant difference between the accuracies of the US methods in terms of PNP diagnosis. What was the agreement between the two methods for PNP diagnosis? If possible, statistical analysis should be provided including the statistical tests that were used in data analysis. The specific software used for statistical analyses should also be mentioned. If statistical analysis was not possible, this should be further explained.
The statistical analysis has been specified in the text.
The diagnostic accuracy of the three different methods has been calculated and compared (CT: 99%, C-mode US: 97%, B-mode US: 81%. Data has been insert in the Results.
The inter-observer concordance index for the C-mode between R1 and R2 has been also calculated with value of 99% (Cohen’s k: 0.979); the same index for B-mode was 99% (Cohen’s k: 0.946). The test used was Cohen’s k test.
As suggested, we have calculated the agreement between the two US methods for PNP diagnosis in R1 and R2. The R1 intra-observer concordance index for C-mode and B-mode detection of PNP was 77%, but the Choen’s k was scarce (0.175); the R2 intra-observer concordance index for C-mode and B-mode detection of PNP was 74% with modest Choen’s k (0.253). Data have been interpreted as confirm of limited validity of B-mode US as diagnostic test in air detection with performance significantly inferior to C-mode.
Therefore, to improve statistical information, as recommended, a dedicated Table has been added (Table 2).
- Is this the first study addressing the utility of this US contrast-specific software in the diagnosis of PNP or are there any relevant data found in the literature? This should be mentioned and analysed in the discussion section.
You are absolutely right. To date, we do not found similar studies in literature; we have added this mention in the discussion.
Minor comments
- In my opinion, the introduction section seems somewhat long.
The introduction has been shortened.
2. A table showing the sensitivity, specificity, PPV, NPV, and accuracy for each diagnostic method would be useful.
A table has been added (Table 2).

Round 2
Reviewer 2 Report
All my points were adressed satisfactorily. I have no further complaints.
Reviewer 3 Report
I have no comments